# Recent Progress and Future Prospect of CRISPR/Cas-Derived Transcription Activation (CRISPRa) System in Plants

**DOI:** 10.3390/cells11193045

**Published:** 2022-09-28

**Authors:** Xiao Ding, Lu Yu, Luo Chen, Yujie Li, Jinlun Zhang, Hanyan Sheng, Zhengwei Ren, Yunlong Li, Xiaohan Yu, Shuangxia Jin, Jinglin Cao

**Affiliations:** 1Institute of Cotton, Shanxi Agricultural University, Yuncheng 044000, China; 2Hubei Hongshan Laboratory, National Key Laboratory of Crop Genetic Improvement, College of Plant Science and Technology, Huazhong Agricultural University, Wuhan 430070, China; 3Tobacco Research Institute of Hubei Province, Wuhan 430030, China

**Keywords:** CRISPRa, CRISPR/Cas, dCas9, genome editing, transcription activation

## Abstract

Genome editing technology has become one of the hottest research areas in recent years. Among diverse genome editing tools, the Clustered Regularly Interspaced Short Palindromic Repeats/CRISPR-associated proteins system (CRISPR/Cas system) has exhibited the obvious advantages of specificity, simplicity, and flexibility over any previous genome editing system. In addition, the emergence of Cas9 mutants, such as dCas9 (dead Cas9), which lost its endonuclease activity but maintains DNA recognition activity with the guide RNA, provides powerful genetic manipulation tools. In particular, combining the dCas9 protein and transcriptional activator to achieve specific regulation of gene expression has made important contributions to biotechnology in medical research as well as agriculture. CRISPR/dCas9 activation (CRISPRa) can increase the transcription of endogenous genes. Overexpression of foreign genes by traditional transgenic technology in plant cells is the routine method to verify gene function by elevating genes transcription. One of the main limitations of the overexpression is the vector capacity constraint that makes it difficult to express multiple genes using the typical Ti plasmid vectors from Agrobacterium. The CRISPRa system can overcome these limitations of the traditional gene overexpression method and achieve multiple gene activation by simply designating several guide RNAs in one vector. This review summarizes the latest progress based on the development of CRISPRa systems, including SunTag, dCas9-VPR, dCas9-TV, scRNA, SAM, and CRISPR-Act and their applications in plants. Furthermore, limitations, challenges of current CRISPRa systems and future prospective applications are also discussed.

## 1. Introduction

Gene expression involves multiple processes, including transcription of DNA into messenger RNA (mRNA), splicing of mRNA, translation, and post-translation modification. Accurate regulation of DNA transcription into mRNA is the first step to control the complex process of gene expression. Directional regulation of gene expression will contribute to our understanding of cell physiology, and it is essential for advances in biotechnology.

For the regulation of endogenous gene expression, manipulating transcription factors (TFs) to target the specific target gene promoters to activate/inhibit gene transcription is the classical strategy and a successful one [1]. For example, in mammals, simultaneous up-regulation of four transcription factors reversed differential cells to pluripotent stem cells. However, it is difficult to regulate multiple genes’ transcription due to the specificity of TFs binding sites at DNA [2]. Researchers circumvented these limitations by designing a TFs binding site for promoters to regulate target genes transcription [3]. However, naturally occurring TFs have extensive DNA binding activity, which limits the specificity and efficiency of this method.

Transcription factors can only bind to fixed sites and lack flexibility. Site-specific nucleases (SSNs) such as TALEN and CRISPR/Cas9 have emerged as multipurpose tools which can greatly enhance molecular biologists’ capability such as knock out, base edit, knock in, knock up and knock down the target gene [4,5,6,7]. These SSN systems emerged as genome editing tools, and introduce DNA double-strand breaks (DSBs) anywhere in a particular genome [8,9,10]. In known DNA binding modules, inactivation of Cas9 (dead Cas9, dCas9) fused to transcriptional activators or repressor [9,11] can effectively regulate multiple genes’ transcription under the guidance of different gRNAs [12]. Overexpression of foreign genes by traditional transgenic technology in plant cells is the routine method to verify gene function and shape gene regulation. However, there are still some limitations for the wide application of this strategy. One of the main limitations is the vector capacity constraint that makes it difficult to express multiple genes using the typical Ti plasmid vectors from Agrobacterium. The cumbersome of stacking gene cloning protocol is another obstacle that limits its application. The recent advancements in CRISPR-based gene activation have offered powerful and specific induction of gene expression that overcome the limitations of traditional gene overexpression methods. Multi genes activation can be effectively realized through the CRISPRa system which provides more possibilities for the application of plant genetic improvement in the future.

In this paper, we summarized the emergence and development of transcriptional activation system based on CRISPR/dCas9 and its application in plant research. To provide a reference for plant researchers to compare the differences of different activation systems and to regulate the activation efficiency of endogenous genes in plants in the future. In addition, the potential applications, existing problems, and challenges for these new technologies were also discussed.

## 2. Composition of the CRISPR/Cas-Derived Activation System

### 2.1. CRISPR/Cas System

The CRISPR/Cas system was first investigated in 1987: scientists discovered a kind of unique DNA sequences from *Escherichia coli* genome, which are near the *iap* gene sequence and were called Clustered Regularly Interspaced Palindromic Repeats (CRISPR). However, its biological significance was unclear at that time [13]. Subsequent studies showed that CRISPR/Cas is a complex adaptive defense mechanism in prokaryotes against invading viruses or plasmid DNA [14]. Researchers found that about 40% bacteria and about 90% archaea are present in this powerful defense system [15]. Cas proteins involved in this defense mechanism have also been identified [16].

According to the repeat sequence identity of CRISPR and their Cas protein sequence homology, CRISPR/Cas system was classified into two classes and six types (as shown in Figure 1) [17,18,19]. Class I CRISPR/Cas systems require large effector protein complexes, which are classified into Type I, Type III, and Type IV. Class II CRISPR/Cas systems are classified into Type II, Type V, and Type VI, requiring only an RNA-directed endocytase to cut invading genetic components. The simplicity and efficiency of Class II system make it work as widely used genome editing tool. Type II CRISPR/Cas systems have been used in a variety of organisms, including microbes [20], fungi [21], animals [22], and plants [23].

### 2.2. Cas9 and dCas9

Cas9 is a specific DNA endonuclease that existed in bacteria species such as *Streptococcus sepsis*, *Staphylococcus aureus*, and *Streptococcus thermophilus*. It is a multifunctional protein with two ribozyme domains HNH and RuvC. First, Cas9 forms a ribosome protein complex with two small non-coding RNAs, CRISPR RNAs (crRNAs) and trans activated crRNAs (tracrRNAs). Then the RNA complex will find and identify a suitable Protospacer Adjacent Motif (PAM), such as the 5‘-NGG-3’ sequence of the target sequence [24]. Finally, Cas9 with RNA complex search and match the crRNA target sequence, and then the HNH ribozyme domain will cut the target chain, while the RuvC domain will cut the reverse chain subsequently [25,26,27].

Qi et al. (2013) mutated two conserved endonuclease domains of Cas9 in CRISPR/Cas9 system. The aspartic acid at position 10 of RuvC domain was mutated to alanine (D10A) and histidine at position 840 of HNH domain was mutated to alanine (H840A), so that Cas9 protein lost endonuclease activity and became dead Cas9 (dCas9), which cannot cut DNA but still can bind to specific target DNA sequences with the guide RNA [28]. Therefore, dCas9 binds to the upstream region of the promoter transcription start site (TSS) region and disrupts RNA polymerase or transcription factor binding to the promoter, resulting in inhibition of gene expression without altering the genome. This gene transcription regulation strategy was defined as CRISPR interference (CRISPRi) [29]. The inhibition intensity of CRISPRi system can reach 1000-fold mostly in prokaryotes [30]. CRISPR interference is also widely used in plants such as *Nicotiana tabacum*, *Zea mays*, and *Arabidopsis thaliana* (as shown in Table 1) [31,32,33,34,35,36]. CRISPR interference can be used as an alternative tool for RNAi. Conversely, CRISPR/Cas-derived transcriptional activation (CRISPRa) also can be achieved by dCas9. Bikard et al. (2013) fused the dCas9 protein with ω subgroups (*rpoZ*) in *E. coli*, and this dCas9-ω complex increased the reporter gene transcriptional level up to 2.8-fold [37]. The models of CRISPR mediated transcriptional regulation are shown in Figure 2.

### 2.3. Guide RNA (gRNA)

Guide RNA (gRNA) of CRISPR/Cas9 system is a specific RNA sequence composed of two elements: crRNA and tracrRNA. The gRNA recognizes the target DNA and directs Cas9 protein to produce double-strand breaks (DSB) in target DNA [10]. Any complementary gene or nucleotide of sgRNA sequences can be targeted by CRISPR systems. Furthermore, Cas9 and dCas9 can use multiple gRNAs effectively to expand the flexibility and multiplicity of CRISPR/Cas system by editing several different target genes simultaneously [38]. The selection of different regulatory elements and the change in regulatory efficiency can be achieved through the modification of scaffolds [39].

### 2.4. Transcriptional Regulators

Transcription regulatory factors are essentially chimeric proteins, and their DNA binding domains are connected with the functional domains that control the transcription mechanism by promoting the recruitment of key cofactors to regulate transcription [35]. In the CRISPR/dCas9 meditated transcription regulation system, transcriptional regulation of target genes is achieved by fusing transcription regulators with dCas9. Transcriptional Repression Domains (TRD) include transcription repressors such as the Krüppel-Associated Box (KRAB) domain. Its function is to suppress transcription by collecting co-inhibitors KRAB-Associated Protein 1 (KAP-1), which leads to the formation of heterochromatin complexes that ultimately lead to gene silencing [40]. The transcriptional repressor SRDX originate from Ethylene-responsive element binding factor-associated Amphiphilic Repression (EAR) transcriptional repressor domain, which are effective plant transcriptional repressors [41].

On the other hand, transcriptional activators such as herpes simplex Virus Protein 16 (VP16) Transcriptional Activator Domain (TAD) or tetrameric repeat VP64 [42] could elevate the target genes’ transcription. Plant also has its own specific transcription regulators such as Ethylene Responsive Factor/Ethylene-Responsive Element Binding Proteins (ERF/EREBP). They can maintain high activity even in the presence of activating elements (such as VP16). This domain has been used as a tool for transcriptional inhibition in some studies [40]. Ethylene response factors in the ERF/EREBP family play a leading role in the response to biological and abiotic stress [43]. These transcriptional regulators bind to the APETALA2(AP2) DNA domain and various unnamed motifs [44,45,46]. These sequences are EDLL short sequences consisting of conserved glutamate (E), aspartic acid (D), and leucine (L). Several studies showed that the EDLL motif is a powerful tool for endogenous gene transcriptional activation [47,48]. Transcriptional activation domains and their detail description are summarized in Table 2.

## 3. Strategies for Achieving Transcriptional Activation using the CRISPR/dCas9 System

As mentioned previously, transcriptional activators are essential for the regulation of transcriptional activation through dCas9. Regulation of target gene activation can be achieved by fusing transcriptional activators with dCas9, or by directing gRNA into scaffolds to recruit transcriptional activators. A comparison of different activation systems using these two strategies and evaluating their efficiency is summarized below:

### 3.1. Fusion of Transcriptional Activation Effectors with dCas9

Studies showed that the dCas9 protein fused with the transcription factor VP64 can recruit and stabilize the promoter complex to form dCas9-VP64 [49]. However, dCas9-VP64 is a low-intensity activator with less than 10-fold target gene transcription [32,34,36]. Although higher levels of expression may be desirable to observe more pronounced phenotypes associated with the function of certain genes, these results clearly show that Cas9-based transcription factors can activate endogenous gene expression [50]. Based on this, the researchers attempted to string other different activators together with dCas9 and this resulted in the following three systems:

#### 3.1.1. dCas9- SunTag System

Researchers found that the combination of multiple transcription factors with a single promoter significantly enhanced transcriptional activation of the downstream gene. This principle of signal amplification through protein polymers had been applied in biological system imaging and engineering design [51]. Based on this principle, Tanenbaum et al. (2014) developed a novel protein scaffold, a repeating peptide array called SunTag for transcription activation [52].

In the Super Nova Tag (SunTag) system, dCas9 is fused with tandem General Control Nonderepressible 4(GCN4) peptide repeats and each repeat connects a transcription regulator via an anti-GCN4 antibody called Single Chain Fragment Variable (scFv). The domain has highly affinity with relatively short nucleic acid sequences, allowing for protein polymerization on a single RNA template. Compared to the dCas9-VP64, SunTag enables multiple transcription factors such as Ten-Eleven Translocation (TET) and regulatory elements X to be recruited in one system. Therefore, multiple genes can be synergistically activated by different transcriptional regulators in tandem [53] as shown in Figure 3A.

#### 3.1.2. dCas9-VPR System

In natural gene regulation system in cells, many transcription factor Activation Domains (ADs) can make changes in transcription through a coordinated collection of necessary activators. Chavez et al. (2015) hypothesized that transcriptional activation could be enhanced by combining multiple Ads [54]. A series of more than 20 known transcriptional effectors were fused to the C-terminus of dCas9 in an effort to enhance the transcriptional activation efficiency in human HEK 293T cells. The result show that dCas9-VP64, dCas9-p65 [55] and dCas9-Rta [56] were active. Then researchers used Cas9-VP64 as their starting scaffold and expanded up-regulation with p65 and Rta at the C-terminal, resulting in VP64-p65-Rta (VPR) activator complex as shown in Figure 3B. This novel activator displayed increased transcription levels of endogenous target gene, ranging from 20-fold to 320-fold over the original dCas9-VP64 activator [54].

#### 3.1.3. dCas9-TV System

Currently, the CRISPRa system is continually being optimized and improved in animal cells, whereas it is still at an early stage for plant cells. The dCas9-TV system is the first CRISPR/dCas9 activation system applied in plant cells [57].

Li et al. (2017) modified dCas9-VP64 with VP16’s octahedron VP128 instead of VP64 and this system activated *LUC* with a five-fold increase; an improved performance on the dCas9-VP64 (only a two-fold increase). TAD was then introduced as a second step to enhance dCas9-VP128 activity, including plant-specific EDLL and modified ERF2 (ERF2m) and TAD from the herpes simplex virus’ TALE element. Results showed that the combination of VP128 with a tandem ERF2m-EDLL motif (up to four copies) activated *LUC* transcription up to 12.6-fold compared to the base level, while combination of VP128 and TALE TAD (up to six copies) increased *LUC* transcription up to 55-fold. The results suggested that dCas9-6TAL-VP128 was a strong transcriptional activator and was called the dCas9-TV system [39,57] as shown in Figure 3C.

### 3.2. Modification of gRNA into a Scaffold and Recruitment of Transcriptional Activators

A common method for studying RNA localization is to insert multiple MS2-binding RNA scaffolds into the target RNA molecule and then tag the RNA molecule with GFP by recruiting MS2-GFP fusion proteins [51]. The inherently modular and programmable nature of RNA allows it to be used to coordinate biological assembly. First of all, RNA can recognize DNA targets via the complementary base pairing principle. Second, RNA contains a domain of RNA-protein interactions that are useful for recruiting specific proteins. Previous studies demonstrated that RNA scaffolds can coordinate the assembly of functional proteins [58]. Therefore, a second common strategy for CRISPR/dCas9 derived transcriptional activation system is to use gRNA as scaffold to recruit transcriptional activators.

#### 3.2.1. scRNA System

Zalatan et al. (2015) constructed a scaffold RNA (scRNA) system and demonstrated that it can effectively activate gene transcription in yeast and human cells [29]. ScRNA systems are inspired by natural regulatory systems in which scaffold proteins physically assemble interacting components of cell signaling pathways. Similar scaffolding principles were applied in genomic modification, such as using Long-strand Non-Coding RNA (lncRNA) as assembly scaffolds to recruit key epigenetic modifiers at specific genomic sites [59,60]. In this system, the researchers fused the well-characterized viral RNA sequences including MS2, PP7, and Com into the 3′-end of gRNA, which are recognized by RNA binding proteins, respectively. Then, the transcriptional activation domain VP64 was fused to each corresponding RNA binding proteins, as shown in Figure 3D.

#### 3.2.2. SAM System

Synergistic Activation Mediator (SAM) continues to be constructed by modifying gRNA loops. Initially, the researchers tried to find the best anchoring position for the activation domain in the Cas9-sgRNA complex. Previously, dCas9-based transcription activators relied on trans activation domains fused to the N or C ends of dCas9 proteins. For an investigation regarding alternative anchorage sites, Konermann et al. (2015) investigated the crystal structure of dCas9 and revealed that the distal gRNA loops did not interact with dCas9 at all. Other research revealed that the gRNA loops could fuse with protein-interacting RNA domains to promote transcription factor recruitment to dCas9 [61]. They combined VP64 with the NF-κB activation p65 [62] and reintroduced the activation domain of Human Heat Shock Transcription Factor 1 (HSF1) for improving the efficiency of dCas9-mediated gene activation (Figure 3E). Finalization of MS2-p65-HSF1 fusion protein improved *ASCL1* (12%) and *MYOD1* (37%) transcription activation [61]. Some results demonstrated that the SAM gene activation platform can facilitate in vivo research and drug discovery [63].

#### 3.2.3. CRISPR-Act 2.0 System

The CRISPR-Act 2.0 system is an improved system for multiple transcriptional activation in plants, which is also dependent on MS2. It is capable of recruiting four VP64 proteins to gRNA and the fifth VP64 was carried by the dCas9-VP64 complex. As shown in Figure 3F, this system can carry a total of five activators to the target site. When compared to the original dCas9-VP64 system, CRISPR-Act 2.0 increased transcriptional activation efficiency by three to four folds [64]. Additionally, the CRISPR-Act 2.0 system enables simultaneous activation of multiple genes in vivo. Lowder et al. (2017) assessed the efficacy of the CRISPR-Act 2.0 system in rice. They simultaneously activated three independent endogenous genes *Os11g35410*, *Os03g01240* and *Os04g39780* in rice protoplasm [48,64]. The results indicated that CRISPR-Act 2.0 was more effective than the previous dCas9-VP64 systems.

#### 3.2.4. CRISPR-Act3.0 System

Based on the CRISPR-Act 2.0 system, Pan et al. (2021) combined dCas9-VP64, gR 2.0 scaffolds, 10xGCN4 SunTag and a newly developed 2xTAD activator to build a novel CRISPR-activating system, CRISPR-Act3.0 [65], as shown in Figure 3G. Multigene activation was achieved by assembling gRNA with multiple activators. At the same time, the further combination with CRISPR-Cas12b and the SpCas9 variant SPRY may expand the target range of CRISPR-activation. The primary objective of functional genomics is to determine the causal relationship between gene expression and phenotypic characteristics. By regulating gene expression in plants, the CRISPRa system provides a novel, efficient method to simplify and accelerate these studies. The future holds a lot of potential for improving CRISPR-activation efficiency, flexibility, and scalability.

## 4. Application and Limitation of CRISPRa System in Plants

First, the researchers used *A. thaliana* and *N. benthamiana* for transcription activation test to evaluate the transcriptional activation activity of dcas9-VP64. The data suggested that dCas9-VP64 had weak activation when a single sgRNA was designed in the vector. Then, the adoption of multiple sgRNAs to target the same gene promoter resulted in higher level of gene activation [32,35,47,66]. Transcription activation domains from plant AP2 transcription factors as known as the EDLL motif and bacterial TALE protein have been used to construct dCas9-based transcriptional activators in plants [64,66]. Li et al. (2017) designed a transcriptional activation system dCas9-TV that works effectively in plants. High-level transcriptional activation of target genes in *A. thaliana* and *O. sativa* and rice cells was achieved with a 201-fold and 2745-fold increase, respectively [57]. Xiong et al. (2021) further optimized dCas9-TV system in rice to increase the transcription of *OSER1* gene up to 4000-fold [67]. Researchers also tried to link the EDLL motif with VP64, in a similar fashion to the VPR strategy, in order to increase the efficiency of transcriptional activation in rice [68]. Selma et al. (2022) used multipliable CRISPR activator dCasEV2.1 activated 3 and 7 genes with gene activation levels ranging from 4- to 1500-fold in *N. benthamiana* [69].

Similarly, transcriptional activation of endogenous target genes was also efficiently applied in *A. thaliana* when SAM or SunTag were fused with the dCas9 [70,71]. When using the same gRNA, the CRISPR-Act2.0 system significantly outperformed the dCas9-VP64 system to activate target genes in *A. thaliana* cells with a 1500-fold increase [64]. CRISPR-Act3.0 recruits additional activators based on the SunTag system coupled to MS2-MCP which increased activation efficiency of endogenous genes in rice (250-fold), *Arabidopsis* (4000-fold) and tomato (240-fold), respectively [65]. It is important to note that even with effective activation systems, such as CRISPR-Act 2.0 and dCas9-TV, some endogenous genes are also difficult to activate effectively [64,66]. It is speculated that dCas9 and activators must compete with endogenous transcription activators to bind the specific region of promoters. The activation system and activation efficiency applied in plant species are summarized in Table 2.

The current CRISPR/dCas9-based transcription system needs to be delivered in plant by *Agrobacterium*, Polyethylene Glycol (PEG) or particle bombardment techniques. The donor plants should harbor the T-DNA insertion with CRISPR/dCas9 fragment may be regulated under the biosafety framework for GM plants since they contain transgene element in the genome. This is maybe the major concern for wide application of the current CRISPR/dCas9 mediated transcription activation system in plant. In addition, with the miniaturization of Cas protein and the maturity of virus introduction technology, it will leave the tissue culture stage and accelerate the efficiency of related research. It is anticipated that transcriptional regulation using CRISPR will be further improved by overcoming the technical difficulties mentioned above in the near future.

## 5. Concluding Remarks and Perspective of Transcription Activation System in Plants

This paper reviews the different strategies developed based on CRISPR/dcas9 activation system in recent years. These strategies have been widely employed to study gene transcription regulation in animal cells; however, their application in plants is still at its early stages. In plants, this system is mainly used in the model plant species such as Arabidopsis and rice, but not in other major crop plant species. Nevertheless, there is still a great deal of potential for research in this field. First, the multi gene activation system enables large-scale transcriptional regulation in plants in order to better understand gene regulatory networks. Second, the up-regulation of multiple key genes in the metabolic pathway can generally result in the production of valuable commercial products, and synthetic biology is likely to make a significant breakthrough in the field of agriculture [72]. Additionally, the improvement of the protein complexes function is mainly dependent upon the simultaneous activation of many different genes, and the multigene activation system is helpful for studying the functional region of complex proteins [73]. Finally, directed activation of multiple defense genes against pathogen attack is a potential strategy to improve plant immunity without affecting traits [74]. Plants are able to recognize insect molecules and respond accurately and defensively when insects are grazing. However, the effectors released by insects interfere with the host plant’s defensive response. We can use the CRISPR-activation system to avoid the destruction of plant defense mechanisms by insect effectors and create broad-spectrum insect-resistant plant materials. Several novel gene editing tools such as Cas12a T-containing PAM with short guide RNA (42–44 nucleotides of crRNA), appeared to be more effective in regulating multiple genes [75].

Ongoing efforts are being made to update the transcriptional activation system based on CRISPR/dCas9. For example, Chiarella et al. (2020) improved CRISPR/dCas9-based transcriptional activation using Chemogenetic Epigenetic Modifiers (CEMs). By employing endogenous chromatin activators, this system was able to activate target gene expression without requiring exogenous transcriptional activators, leading to dose-dependent activations of target genes [76]. Gamboa et al. (2020) integrates the heat shock switch with dCas9 complex to remotely control gene activation and inhibition with short-time heating pulses [77]. The experimental results demonstrated that the activation intensity of the dCas9-VP64 complex with heat as remote trigger depends on thermal pulse and can be substantially improved in only 15 min. With this heat-activated transcriptional activation system, CRISPR/dCas9 can activate transcription without invasive procedures. Optical regulation can be used as another new method for inducing and regulating endogenous genes in plants. Moreover, in an attempt to overcome the problems associated with the compatibility of optogenetic tools with plant growth requirements, Ochoa-Fernandez et al. (2020) developed Plant-Usable Light-Switch Elements (PULSE). A combination of blue light sensing inhibition regulation with red light sensing activation regulation resulted in gene expression being regulated only in the presence of red light. Combining PULSE with CRISPR/dCas9 mediated gene activation system (dCas9-TV) demonstrated light controlled activation of *A. thaliana* [78].

As opposed to ribozymes with cutting activity, which often cause uncertain genomic modifications and result in chromosome rearrangements or deletion during multiple point editing, dCas ribozymes offer a superior solution to these problems [79]. However, the challenge remains in creating long sequences of multiple gRNAs strung together and determining their editing efficiency for various targets [80]. Under a multilocus CRISPR editing system, the increased number of gRNAs leads to limited dcas9 competition between different gRNAs, which in turn leads to variations in target gene editing efficiency [81], as well as uncertainty in the regulation of target genes by gRNAs [82]. As mentioned previously, CRISPR/dCas9 derived transcription activation system exhibited obvious advantages over the traditional overexpression strategy to elevate the target genes’ transcription.

Epigenetic regulation is an important way to regulate gene expression and has certain reversibility. The researchers used CRISPR/Cas9 to knock out epigenetic factors to determine the role of epigenetic factors in the regulation of endogenous genes in plants [83]. In addition, researchers can combine CRISPR/dCas with different epigenetic effector domains for specific epigenetic regulation of target sites [84]. In the future, with the improvement of episequencing technology and the in-depth study of epigenetic regulators, CRISPR-based epigenetic regulation research will have room for development.

Finally, with the wide application of single-cell sequencing technology in plants [85], it will further improve the transcriptional regulation information of plants [86] and open up new space for the application of CRISPR-activation system in plants.

## Figures and Tables

**Figure 1 cells-11-03045-f001:**
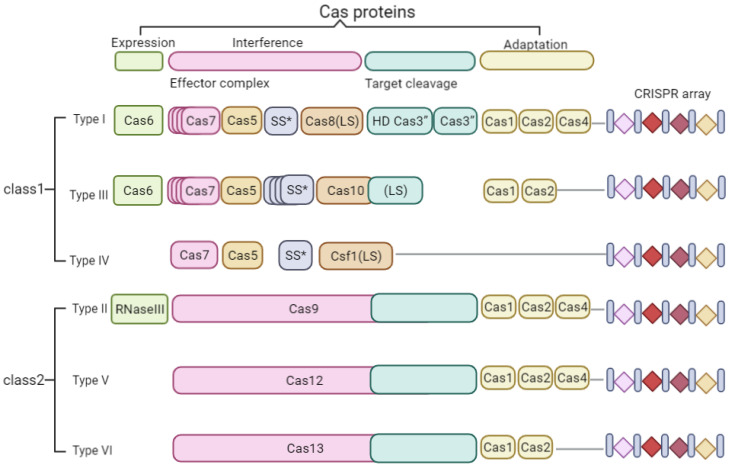
Characteristics of different types of CRISPR/Cas systems. CRISPR/Cas systems are classified as types I to VI. Type I systems are characterized based on the occurrence of signature protein Cas3, a protein which contains both DNase and helicase domains used to degrade the target. Type II CRISPR/Cas systems use Cas1, Cas2, Cas9, and a fourth protein (Csn2 or Cas4), whereas the type III CRISPR/Cas systems comprise the Cas10 with an indistinct role. The type II CRISPR/Cas system originates from *S. pyogenes* and comprises three components: the CRISPR RNA (crRNA), trans-activating crRNA (tracrRNA), and a Cas9 protein. The type V CRISPR/Cas system (Cas12) is an RNA-guided system which is analogous to CRISPR/Cas9 but exhibits some unique characteristics. This CRISPR system relies on a T-rich sequence at the 5′-end of the protospacer sequence (5′-TTTN-3′ or 5′-TTTV-3′; V = A, C, or G, in some cases), as opposed to the G-rich, NGG sequence for Cas9. The type VI system (Cas13) is effector protein for RNA cutting, which is used as RNA-guided ribonuclease, the nonspecific, trans-acting RNase activity of which is activated by base pairing of the crRNA guide to an ssRNA target. The Cas7, Cas5,SS*, Cas8(LS), Cas10 and CSf1(LS) have been drew with different colors, but they all belong to interference part of class I.

**Figure 2 cells-11-03045-f002:**
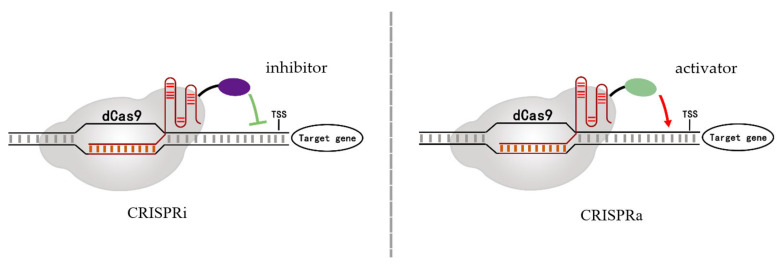
The diagram of CRISPR/dCas9-mediated transcriptional regulation. The dCas9 fused with transcriptional inhibitors or activators can provide additional inhibition (CRISPRi) or activation (CRISPRa) functions.

**Figure 3 cells-11-03045-f003:**
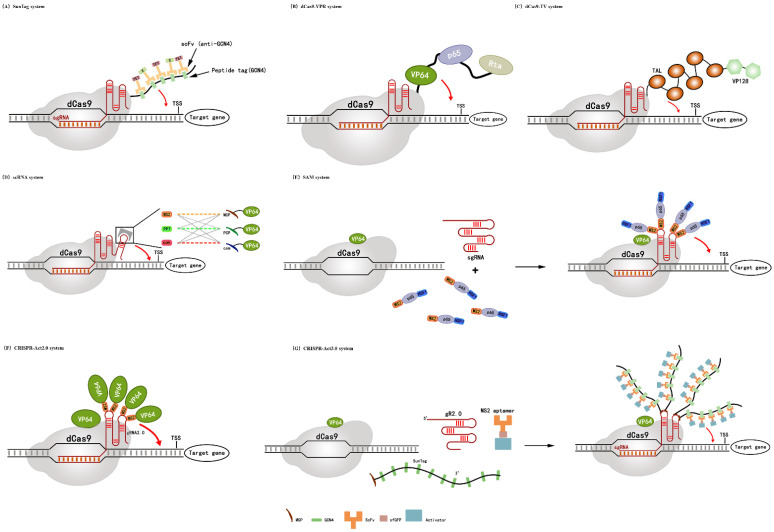
Schematic diagram of CRISPR transcriptional regulation. (**A**) SunTag system: dCas9 fused with GCN4 to recruit multiple copies of scFv, TET and other element X to activate the target gene cooperatively; (**B**) dCas9-VPR system: dCas9 fused with VP64-p65-Rta to activate the target gene; (**C**) dCas9-TV system: dCas9 TV activation system includes 6 TAL and 2 VP128; (**D**) scRNA system: An RNA hairpin domain with RNA sequences MS2, PP7 and com recognized by MCP, PCP and com RNA binding proteins was introduced at the end of sgRNA, and the transcription activating element VP64 was fused into each corresponding RNA binding protein; (**E**) SAM system: The four bases at the distal end of the stem loops of gRNA were modified to recognize the stem loops of MS2 to bind p65 and HSF1; (**F**) CRISPR-Act2.0 system: dCas9 was fused with VP64 and the four bases at the distal end of the stem loops of gRNA were modified to recognize the stem loops of MS2 to bind VP64; (**G**) CRISPR-Act3.0 system: SunTag system with the MS2–MCP interaction would recruit more activator.

**Table 1 cells-11-03045-t001:** Applications of CRISPR interference in plants.

Repressor	Plant Species	Target Gene	Highest Repression Level (%)	References
dCas9	*N. benthamiana*	*PDS*	20	(Piatek et al., 2015)
		*pNOS::LUC* reporter	80	(Vazquez-Vilar et al., 2016)
dCas9-SRDX	*N. benthamiana*	*PDS*	33	(Piatek et al., 2015)
		*pNOS::LUC* reporter	50	(Vazquez-Vilar et al., 2016)
	*Z. mays*	*ChlH*	75	(Irene et al., 2020)
	*Z. mays*	*PDS*	60	
dCas9-BRD	*N. benthamiana*	*pNOS::LUC* reporter	60	(Vazquez-Vilar et al., 2016)
dCas9-3 × SRDX	*A. thaliana*	*CSTF64*	60	(Lowder et al., 2015)
		*miR159a*	80	
		*miR159b*	70	
dLbCpf1-SRDX	*A. thaliana*	*miR159b*	90	(Tang et al., 2017)
dAsCpf1-SRDX	*A. thaliana*	*miR159b*	90	

**Table 2 cells-11-03045-t002:** Applications of the CRISPR-activation system in plants.

Activator	Plant Species	Target Gene	Fold Change (Highest Level)	References
dCas9-EDLL	*N. benthamiana*	*NbPDS*	3.5	(Piatek et al., 2015)
		*pNOS::LUC* reporter	2.2	(Vazquez-vilar et al., 2016)
dCas9-TAL	*N. benthamiana*	*AtPDS*	4	(Piatek et al., 2015)
dCas9-VP64	*A.* *thaliana*	*AtPAP1*	7	(Lowder et al., 2015)
		*miR319*	7.5	
		*AtFIS2*	400	
	*N. benthamiana*	*pNOS::LUC* reporter	2.3	(Vazquez-vilar et al., 2016)
	*A.* *thaliana*	*AtWRKY30*	2.1	(Li et al., 2017)
		*AtRLP23*	0.9	
		*AtCDG1*	4.3	
	*O. sativa*	*OsGW7*	2.7	
		*OsER1*	0.3	
		*Os03g01240*	2.1	(Lowder et al., 2018)
		*Os04g39780*	1.1	
		*Os11g35410*	2.2	
	*Z. mays*	*PDS*	2.5	(Irene et al., 2020)
		*TrxH*	2.0	
dCas9-VP64 + MS2-p65-HSF1 (SAM)	*A.* *thaliana*	*AtAVP1*	5	(Park et al., 2017)
		*AtPAP1*	7	
dCas9-4 × EE-2 × VP64	*A.* *thaliana*	*pWRKY30::LUC* reporter	12.6	(Li et al., 2017)
dCas9-6 × TAL-2 × VP64 (dCas9-TV)	*A.* *thaliana*	*AtWRKY30*	138.8	(Li et al., 2017)
		*AtRLP23*	32.3	
		*AtCDG1*	92.2	
	*Oryza sativa*	*OsGW7*	78.8	
		*OsER1*	62	
dCpf1-TV	*A.* *thaliana*	*pWRKY30::LUC* reporter	4.7	(Li et al., 2017)
dCas9-VP64-EDLL	*A.* *thaliana*	*AtPAP1*	4	(Lowder et al., 2018)
		*AtFIS2*	3	
	*O. sativa*	*OsCGA1*	5	(Lee et al., 2021)
dCas9-VP64 + MS2-EDLL	*A.* *thaliana*	*AtPAP1*	30	(Lowder et al., 2018)
		*AtFIS2*	30	
dCas9-VP64 + MS2-VP64 (CRISPR-Act2.0)	*A.* *thaliana*	*AtPAP1*	45	(Lowder et al., 2018)
		*AtFIS2*	1500	
		*AtULC1*	40	
		*miR319*	6	
	*O. sativa*	*Os03g01240*	3	
		*Os04g39780*	4	
		*Os11g35410*	2.8	
dCas9-2 × GCN4 + scFv-sfGFP-VP64 (SunTag)	*A.* *thaliana*	*AtFWA*	140	(Papikian et al., 2019)
		*AtEVD*	4000	
		*AtAP3*	350	
		*AtCLV3*	130	
dCasEV2.1 (EDLL-MS2:VPR/gRNA2.1)	*N. benthamiana*	NbAN2	4000	(Selma et al., 2019)
		NbDFR	10000	
		NbPAL	400	(Selma et al., 2022)
		NbC4H	4	
		Nb4CL	15	
		NbCHS	18000	
		NbCHI	45	
		NbF3H	140	
		NbFLS	40	
dCas9-TV	*O. sativa*	*OsER1* *OsGW7*	4000 200	(Xiong et al., 2021)
	*Gossypium hirsutum*	*Ghpapid*	41.7	(unpublished data)
		*Ghaepsp*	16	
CRISPR-Act3.0 gR2.0 4xGCN4	*O. sativa*	*OsGW7*	45	(Pan et al., 2021)
		*OsER1*	90	
CRISPR-Act3.0 gR2.0 10xGCN4		*OsGW7*	70	
		*OsER1*	95	
CRISPR-Act3.0 gR8xMS2 4xGCN4		*OsGW7*	45	
		*OsER1*	40	
CRISPR-Act3.0 gR8xMS210xGCN4		*OsGW7*	15	
		*OsER1*	10	
CRISPR-Act3.0 VP64 4xGCN4		*OsER1*	30	
CRISPR-Act3.0 VP64 10xGCN4			90	
CRISPR-Act3.0 2xTAD 4xGCN4			140	
CRISPR-Act3.0 2xTAD 10xGCN4			250	
CRISPR-Act3.0 2xTAD–VP64 4xGCN4			120	
CRISPR-Act3.0 2xTAD–VP64 10xGCN4			50	
CRISPR-Act3.0 TV 4xGCN4			35	
CRISPR-Act3.0 TV 10xGCN4			25	
CRISPR-Act3.0 VPR 4xGCN4			30	
CRISPR-Act3.0 VPR 10xGCN4			45	
M-Act3.0 (Multiple sgRNAs)		*OsDXS*	9	
		*OsPDS*	6	
		*OsPSY*	17	
		*OsCRTISO*	3	
		*OsZISO*	23	
		*OsZDS*	3.5	
		*OsCYB*	11	
		*OsCHS*	30	
		*OsCHI*	2	
		*OsF3H*	130	
		*OsDFR*	20	
		*OsLAR*	70	
	*A.* *thaliana*	*AtFT*	240	
		*AtTCL1*	8	
		*AtEVD*	4000	
		*AtAP3*	350	
		*AtCLV3*	130	
	*Lycopersicon esculentum*	*LeSFT*	240	

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
