# Peer review of "Recent Progress and Future Prospect of CRISPR/Cas-Derived Transcription Activation (CRISPRa) System in Plants"

_cells, 2022, doi:10.3390/cells11193045_

Round 1

Reviewer 1 Report

Authors present a review describing the strategies based on CRISPR/dCas9 activation system. Authors start by presenting a description of CRISPR/Cas activation system, focusing on its different components, CRISPR, Cas and transcriptional regulators. Authors also present some works on the application of CRISPRa in plants and focus on the advantages of CRISPRa in agriculture, as well as in other fields.

The subject of the MS is very interesting and it is presented in a way that is easy to read. The MS is mostly well written but there are several aspects that should be revised before it could be accepted for publication.

Lines 20-21: authors mention CRISPR activation system but they could introduce a sentence in the abstract to clearly mention what it is.

Line 27: Please revise first sentence. Gene transcription involves the formation of mRNA. Translation is a further process, maybe authors meant ‘Gene expression involves multiple processes…’?? Please verify. 

Lines 358 until the end: authors might develop a little more. For example authors do not mention viral vectors for CRISPR delivery.

Minor:

There are many places where authors have extra spaces between words, this happens in lines 2, 10, 14, 38, 132, 206, 227, 238, 260, 273, 296, 312, 322, 328, 339, to name a few. Please check.

Line 17: Put a comma (,) between ‘RNA’ and ‘provides’

Line 20: Use ‘summarizes’ instead of ‘will summarize’

Line 21: Use ‘systems’ instead of ‘system’

Line 23: Use ‘are also discussed’ instead of ‘were also discussed’

Line 38: Change ‘though’ for ‘through’

Line 40: Use ‘which limits the specificity’ instead of ‘will limit the specificity’

Line 42: ‘Different with the fixed binding sites for TFs…’ please modify. I understand the meaning but it is confusing as it is.

Line 44: enhance molecular biologists’ capability to what? Please add some information.

Line 56: Use ‘methods’ instead of ‘method’

Lines 56-58: Sentence is confusing, please clarify.

Line 67: Change ‘and so called’ to ‘and were so called’

Line 68: Remove ‘structure’

Line 70: Change ‘against foreign viruses or plasmid DNA invading’ to ‘against invading viruses or plasmid DNA’

Line 72: Change ‘equip’ to ‘present’

Line 73: Use ‘have also been identified’ instead of ‘were identified’

Line 76: Use ‘require’ instead of ‘requiring’

Line 78: Use ‘require’ instead of ‘requiring’

Top of figure 1: ‘Cas proteins’ instead of ‘Cas protions’, please correct figure.

Line 86: Use ‘endonuclease that exist in bacteria’ instead of ‘endonuclease existed in bacteria’

Line 90: Use ‘identify a suitable’ instead of ‘identify suitable’

Line 92: Change ‘match with the crRNA’ to ‘match the crRNA’

Line 92: Change ‘and then HNH ribozyme’ to ‘and then the HNH ribozyme’

Line 100: Change ‘can bind’ to ‘binds’

Line 105: Use ‘plants’ instead of ‘plant’

Line 109: Change ‘is’ to ‘are’

Line 111: Please consider changing the title of table 1 for clarification. Maybe CRISPRi applications in plants.

Table 1: When mentioning Arabidopsis, please include complete species name

Figure 2 should appear near to point 3 of the review, where systems are described. I understand that figure is placed next to first mention (line 109), but it should move to point 3. Authors may also consider creating 2 figures by separating figure 2A from the others.

Line 131: Change ‘Ca9’ to ‘Cas9’

Line 160: Title of table 2 should be ‘Description of transcriptional activation domains’ or similar

Line 167: Change ‘are summarized as following:’ to ‘are summarized below.’

Line 187: Change ‘is highly affinity’ to ‘has high affinity’

Line 193: Remove ‘locally focused’ or clarify meaning

Line 216: Change ‘is’ to ‘was’

Line 216: Change ‘named dCas9-TV’ to ‘was named as dCas9-TV’

Line 279: Change ‘has’ to ‘had’

Lines 291-293: Sentence is confusing, please rephrase

Lines 301-303: Sentence is confusing, please rephrase

Table 3: Please change title to ‘Applications of CRISPR activation system in plants’

Line 311: Use ‘in plants’ instead of ‘in plant’

Line 358: Use ‘needs’ instead of ‘need’

Lines 360-362: Confusing, please clarify.

Reviewer 2 Report

CRISPR/Cas derived Transcription Activation (CRISPRa) System in Plants is nowadays receiving increasing attention as it provides an alternative way to reprogram cellular function without disturbing or damaging the genome.

The topic and scope are quite significant and interesting. The authors described the review potentially.

Authors have added new updates in the CRISPRa system in plants, which is quite helpful to readers working on CRISPR/Cas mediated epigenetic regulation in plants.

The aim of this review article should be elaborated in the introduction section

The authors need to focus more on epigenetic regulation in plants by the CRISPR activation system.

A separate conclusion and future direction section should be provided for the topic instead of merging it with Limitations and perspective of transcription activation system in Plant. Moreover, the conclusion and future direction need to be elaborate.

Data in the tables need to cross-check whether the genes (Tables 1 and 3) mentioned in the tables are edited by CRISPR/Cas system or the CRISPRi or CRISPRa system.

The figures are good, but quality and sharpness need to improve.

Table 1 Statistics of CRISPR interference (CRISPRi) cases in plants need to remove; otherwise, discuss this table in a separate section like 4. CRISPRa system applied in plants.

Table 2. Transcription activator caption needs to revise; write it in detail.

Are epigenetic modifications affect protein structure? If yes, add some information about protein level.

Keywords should be in alphabetical order.

Do not start sentences with abbreviations.

The manuscript can be published after minor revision.

Round 2

Reviewer 1 Report

Authors have clarified all questions raised. I believe the MS can be accepted.

Author Response

Dear editor and reviewers,                                   Sept. 17, 2022

On behalf of all co-authors, we thank you very much for giving us an opportunity to revise this manuscript again. Based on these comments and suggestions, we have made careful modifications on the revised manuscript entitled ‘Recent Progress and Future prospect of CRISPR/Cas -derived Transcription Activation (CRISPRa) System in Plants’ and the authors carefully responded each comment and suggestion.

All authors listed in present manuscript had been connected to discuss and revise this manuscript. The major amendments were made by using the track changes mode in this revised manuscript. We wish that this new version manuscript could meet the journal’s standard.

Point to point responses to editor and reviewers’ comments are as follow:

For the plant species names, full names are used for the first appearance and then the abbreviated name should be used: For example, Arabidopsis thaliana (for the first time), then A. thaliana (for the rest of manuscript). Throughout the text, revisions will be necessary, especially in Tables 1 and 2.

Response: Thanks, as suggested, we modified the species names of plants in the full text and in Tables 1 and 2.

In addition, the reference format in Tables should be corrected: for example, (Piatek 2015) in Table 1 vs. (Li, 2017) in Table 2 [with or without comma]. Probably, (Piatek et al., 2015) and (Li et al., 2017) would be the correct ones, or just use reference numbers.

Response: Thanks, as suggested, we corrected the reference format in Tables such as (Piatek et al., 2015).

Moreover, the tables should be made with compact (by reducing line spacing etc.) and better formats to have a nice look for readers."

Response: Thanks, as suggested, we have reformatted all the tables by reducing line spacing.

Other modifications:

  1. References changed due to new references added.
  2. Other format changes.

Sincerely yours,

Dr. Shuangxia Jin

Full Professor,

Director of Department of crop breeding, College of Plant science and Technology 

National Key Laboratory of Crop Genetic Improvement, Huazhong Agricultural University, Wuhan City, Hubei Province, China

Email: jsx@mail.hzau.edu.cn
